# Wheat GSPs and Processing Quality Are Affected by Irrigation and Nitrogen through Nitrogen Remobilisation

**DOI:** 10.3390/foods12244407

**Published:** 2023-12-07

**Authors:** Yuanxin Shen, Xiaojie Han, Haoxiang Feng, Zhidong Han, Mao Wang, Dongyun Ma, Jianmeng Jin, Shuangjing Li, Geng Ma, Yanfei Zhang, Chenyang Wang

**Affiliations:** 1College of Resources and Environment, Henan Agricultural University, Zhengzhou 450002, China; yxdreamy008@163.com; 2State Key Laboratory of Wheat and Maize Crop Science, College of Agronomy, Henan Agricultural University, Zhengzhou 450046, China; xiaojiehan523@163.com (X.H.); feng13721631928@163.com (H.F.); zhidongh176@163.com (Z.H.); newton.wangmao@163.com (M.W.); xmzxmdy@126.com (D.M.); 17836939676@163.com (S.L.); nymg5135@163.com (G.M.); 3National Engineering Research Center for Wheat, Henan Agricultural University, Zhengzhou 450046, China; 4Crop Research Institute, Kaifeng Academy of Agricultural and Forestry, Kaifeng 475000, China; 15993396527@163.com

**Keywords:** irrigation, nitrogen application, grain storage proteins, winter wheat, nitrogen remobilisation

## Abstract

The rheological properties and end-use qualities of many foods are mainly determined by the types and levels of grain storage proteins (GSPs) in wheat. GSP levels are influenced by various factors, including tillage management, irrigation, and fertiliser application. However, the effects of irrigation and nitrogen on GSPs remain unclear. To address this knowledge gap, a stationary split–split block design experiment was carried out in low- and high-fertility (LF and HF) soil, with the main plots subjected to irrigation treatments (W0, no irrigation; W1, irrigation only during the jointing stage; W2, irrigation twice during both jointing and flowering stages), subplots subjected to nitrogen application treatments (N0, no nitrogen application; N180, 180 kg/ha; N240, 240 kg/ha; N300, 300 kg/ha), and cultivars tested in sub–sub plots (FDC5, the strong-gluten cultivar Fengdecun 5; BN207, the medium-gluten cultivar Bainong 207). The results showed that GSP levels and processing qualities were significantly influenced by nitrogen application (*p* < 0.01), N240 was the optimal nitrogen rate, and the influence of irrigation was dependent on soil fertility. Optimal GSP levels were obtained under W2 treatment at LF conditions, and the content was increased by 17% and 16% for FDC5 and BN207 compared with W0 under N240 treatment, respectively. While the optimal GSP levels were obtained under W1 treatments at HF conditions, and the content was increased by 3% and 21% for FDC5 and BN207 compared with W0 under N240 treatment, respectively. Irrigation and nitrogen application increased the glutenin content by increasing Bx7 and Dy10 levels in FDC5, and by increasing the accumulation of Ax1 and Dx5 in BN207. Gliadins were mainly increased by enhancing α/β-gliadin levels. Correlation analysis indicated that a higher soil nitrate (NO_3_-N) content increased nitrogen remobilisation in leaves. Path analysis showed that Dy10, Dx5, and γ-gliadin largely determined wet glutenin content (WGC), dough stability time (DST), dough water absorption rate (DWR), and sedimentation value (SV). Therefore, appropriate irrigation and nitrogen application can improve nitrogen remobilisation, GSP levels, and processing qualities, thereby improving wheat quality and production.

## 1. Introduction

Wheat is a major grain crop cultivated worldwide, providing essential amino acids, minerals, vitamins, beneficial phytochemicals, and dietary fibre to human diets [1]. Specifically, compared with rice and corn, wheat is a primary source of many types of foods and essential plant-based proteins that depend on the unique characteristics of grain storage proteins (GSPs) [2]. The types and levels of GSPs are associated with the processing and nutritional qualities of wheat. In mature wheat grains, glutenins and gliadins, the main proteins stored in the endosperm, are responsible for the elasticity and extensibility of dough, which in turn determines the processing qualities of various end-products [3]. Glutenins consist of high- and low-molecular weight subunits (HMW-GS and LMW-GS, respectively) and account for 40% of all wheat grain proteins [4]. HMW-GS comprise only 10% of all seed storage proteins but play critical roles in flour-processing operations because they form networks in dough following gluten polymerisation. HMW-GS genes are located at Glu-1 loci on the long arm of group-1 chromosomes (1A, 1B, and 1D) [5]. Many alleles have been discovered at the Glu-A1 and Glu-D1 loci, including one gene encoding an x-type (Ax or Dx) HMW-GS and one gene encoding a y-type (Ay usually silenced or Dy) HMW-GS [6]. LMW-GS content is closely associated with the extensibility and strength of dough [7,8]. Gliadins are classified as ω-, α/β-, and γ-types according to their mobility during acid-polyacrylamide gel electrophoresis, with α-gliadins migrating fastest and ω-gliadins migrating slowest through the gel [9,10].

Global warming and the increasingly scarce water resources are the major problems for wheat production [11]. The composition and relative proportions of glutenins and gliadins are affected by environmental factors [12,13]. Irrigation has a significant impact on wheat yield and quality. Supplemental irrigation at critical wheat growth stages increased crop yield in order to face extreme drought events [11,14]. Flour protein content increased significantly under water deficit, especially total gliadin and glutenin contents, mainly due to the higher nitrogen accumulation and lower carbohydrate accumulation [15], and the upregulated expression of storage protein biosynthesis-related transcription factors Dof and Spa [16]. Irrigation at jointing and flowering stages increased the grain weight [17,18], and irrigation performed at the grain-filling stage resulted in a significant decrease in the grain protein content [19]. When irrigation was performed at ≥180 mm, the gliadin and glutenin contents reduced and caused a decrease in dough strength, and this was mainly contributed by the diluting of carbohydrates to protein content [20,21]. Nitrogen fertiliser plays an important role in balancing yield and processing quality by delaying senescence and increasing nitrogen storage during the flowering stage [22]. The accumulation of grain storage protein is largely influenced by environmental growth conditions during the filling stage, particularly nitrogen availability [23]. Nitrogen application increased the quantities of all glutenin- and gliadin-containing proteins [24] and resulted in the increasing accumulation of GSP [25]. This is mainly due to the upregulated storage protein genes at both transcriptional and translational level [26]. For example, storage protein genes encoding ω-and γ-gliadin and LMW-GS were upregulated at a nitrogen application rate of 150 kg/ha, and the genes encoding α-or β-gliadin were upregulated at a nitrogen application rate of 300 kg/ha [26]. Other research has found that a nitrogen application rate of 225 kg/ha increased the storage protein content by improving the glutamine synthetase activity in wheat grains [27]. Moreover, the genes encoding glutamine synthetase were reported to be regulated by a Dof TF, which showed higher expression levels at high nitrogen application [28]. Previous studies showed that a significant interactive effect of irrigation and nitrogen on wheat quality [29,30]. Wheat grain protein increased with the increasing nitrogen application under drought or low irrigation conditions, while under excessive irrigation, it decreased [29]. The negative effect of irrigation on wheat quality can be reduced to some extent by an appropriate increase in nitrogen application [31].

Most previous studies have focused on the effects of single irrigation or nitrogen on grain yield, nutritional quality, and protein content [32,33,34]. However, little research has focused on the impact of the interaction between irrigation and nitrogen application on GSP, processing quality, and the relationship with nitrogen remobilisation. Therefore, we conducted a long-term irrigation and fertilisation field experiment at two different soil fertility levels. The objectives of the present work were to (1) explore the effects of irrigation and nitrogen application and their interaction on GSP levels and wheat processing quality; (2) clarify the relationships between wheat GSP levels and soil NO_3_-N content/nitrogen remobilisation/processing quality parameters; and (3) determine the optimal combination of irrigation and nitrogen application for GSP levels for two wheat cultivars (*Triticum aestivum* L.) of different gluten strength in two fields with different soil fertility.

## 2. Materials and Methods

### 2.1. Field Experiment and Experimental Design

Long-term stationary field experiments were conducted at Kaifeng Jipo Experimental Farm (34°41′ N, 114°49′ E) where the soil type was clay with high fertility (HF) from 2012, and at Kaifeng Rice Village (34°53′ N, 114°19′ E) where the soil type was sandy with low fertility (LF) from 2014, with a winter wheat/summer maize rotation system utilised in both locations. Field investigations and sampling were performed during the winter wheat growth seasons of 2018–2019 and 2019–2020. The monthly average rainfall and temperature during the 2-year experiments are shown in Appendix A. The physicochemical properties of the 0–20 cm soil layer at the experimental sites were summarised in Table 1.

Two different gluten winter wheat cultivars were used in this experiment; the strong-gluten cultivar Fengdecun5 (FDC5) composed of Ax1, Bx7, By8, Dx5, and Dy10 glutenins, and the medium-gluten cultivar Bainong207 (BN207) composed of Ax1, Bx7, By9, Dx5, and Dy10 glutenins [35]. Gliadins are classified as ω-, α/β-, and γ-types [36]. The sowing periods ranged from 11 to 17 October of each planting season, the sowing amount was 280 kg/ha, and the harvest dates ranged from 26 May to 5 June of the following year.

A split–split block experimental design was employed, with the main plot treated with irrigation, the secondary plot treated with nitrogen fertiliser, and sub–sub plots used to test cultivars (FDC5, the strong-gluten cultivar Fengdecun 5; BN207, the medium-gluten cultivar Bainong 207). Individual plots were 5.8 m × 3 m (17.4 m^2^) for HF and 6.5 m × 3 m (19.5 m^2^) for LF, with three replicates in a randomised design. Three irrigation treatments were performed: no irrigation after sowing (W0), irrigation only during the jointing stage (W1), and irrigation during both the jointing and flowering stages with 75 mm each time (W2). Nitrogen was applied at four levels: 0 kg·hm^−2^ (N0, no nitrogen application), 180 kg/ha (N180), 240 kg/ha (N240), and 300 kg/ha (N300). Fertiliser nutrients were provided in the form of urea (46%), calcium superphosphate (15%) at a rate of 150 kg/ha (P_2_O_5_), and potassium chloride (60%) at a rate of 120 kg/ha (K_2_O). Half of the total urea was used as base fertiliser along with calcium superphosphate and potassium chloride before sowing, and the other half was applied at the jointing stage. The fertiliser application amount was the same in each experimental plot in all years. The other management practices for controlling pests, diseases, and weeds complied with local practices for high-yield wheat production.

### 2.2. Determining the Types and Levels of Glutenins and Gliadins

Grains were stored for post-maturation processing following harvest in 2019–2020, then milled using an FW100 multifunctional pulveriser (Zhongxing, Beijing, China). Glutenins and gliadins were extracted according to a published protocol [37] with minor modifications, and levels were determined by reversed-phase high-performance liquid chromatography (RP-HPLC) using a Waters E2695 + 2998DAD instrument (Waters Corporation, Milford, MA, USA) coupled with a Vydac 218TP C18 chromatographic column (250 mm × 4.6 mm) [38]. HPLC elution gradient parameters for glutenins were 0–10 min with 90% eluent A (0.06% trifluoroacetic acid [TFA] in ddH_2_O) and 10% eluent B (0.05% TFA in acetonitrile), followed by a decrease in eluent A to 35% from 10–65 min [39]. For gliadin elution, eluent A was decreased from 79% to 54% and eluent B was increased from 21% to 46% from 0–51 min, then eluent A was maintained at 79% and eluent B was maintained 21% from 51–58 min. The flow rates for glutenin and gliadin elution were 0.8 mL/min and 1.0 mL/min, respectively, with a column temperature of 60 °C. Samples were filtered through a 0.45 mm nylon membrane and collected in a 2 mL standard sample flask. The injection volume was 10 µL. Component levels were calculated according to the peak area of the corresponding protein spectrum peak based on the different elution times and calculated using the following formula [39]:$$Yu=\frac{Tu\times 100}{M\times {\left(1-X\right)}^{\prime }}$$
where *M* (mg) represents the weight of the sample for extraction; *X* (%) represents the water content of wheat flour measured by an Infratec 1241 near-infrared spectrum analyser (FOSS, Gothenburg, Sweden); *Tu* (AU) represents the peak area of each fraction; 100 accounts for the 10 μL extraction being run on the RP-HPLC system; and the total extraction was 1 mL.

### 2.3. Determining the Nitrogen Content in Organs

Twenty plants were collected from each plot at flowering and maturity stages during the 2018–2019 and 2019–2020 growing seasons. Whole plants were divided into leaves, stems with sheaths, and spikes. Fresh samples were immediately heated in an oven at 105 °C for 40 min then dried at 80 °C until they reached a constant weight. Nitrogen levels were determined by the Kjeldahl method using an K1100F instrument (Haineng, Jinan, Shandong, China). Differences in nitrogen concentration in vegetative organs between flowering and maturity stages (DNC) were calculated as the nitrogen concentration during the flowering stage minus the nitrogen concentration during the maturity stage. The four proteins components (albumins, globulins, gliadins, and glutenins) were separated through sequential extraction by distilled water, 10% NaCl, 70% ethyl alcohol, and 0.5% NaOH, respectively. After adding extraction solution, the mixture was shaken for 30 min at 220 rpm, then centrifuged for 15 min at 4000× *g* Supernatants provided extracts and precipitates were resuspended in the subsequent extraction solution. Each extraction step was repeated three times [40], protein levels in supernatants were measured using the Kjeldahl method with an K1100F instrument (Haineng), and grain proteins were calculated based on the nitrogen content multiplied by 5.7.

### 2.4. Determining the Soil NO_3_-N Content

Mixed soil samples from 0–20 cm soil layers were obtained by soil drilling following harvest in 2018–2019 and 2019–2020 growing seasons. Soil samples were extracted with 2 M KCl solution (1:5 *w*/*v*) by shaking for 1 h. NO_3_^−^ concentration was measured using a continuous flow analyser (Santt System, Skalar, The Netherlands). Soil water content was determined by oven-drying to a constant weight at 105 °C [41].

### 2.5. Determining Flour Quality Traits

Grains were disrupted using a Brabender Junior laboratory mill (Brabender GmbH, Duisburg, Germany) based on approved method 26-21A (AACC, 1995). Wet gluten content (WGC) was measured using a Perten Glutenmatic 2200 gluten testing system (Perkin Elmer, Waltham, MA, USA) and the sedimentation value (SV) was determined by a Zeleny analysis system CAU-B (Shengtai, Jinan, Shandong, China). Dough stability time (DST) and dough water absorption rate (DWR) were determined by a Brabender Farinograph-E instrument (Brabender GmbH).

### 2.6. Statistical Analysis

A two-way MIXED analysis of variance (ANOVA) model was used to examine the interactions between irrigation and nitrogen application on GSPs and processing qualities. For the multiple comparisons, a post hoc Duncan test was employed to compare the differences among the treatments at the *p* ≤ 0.05 and *p* ≤ 0.01 levels by SPSS v.17.0. (SPSS Inc., Chicago, IL, USA). Irrigation and nitrogen applications were taken as fixed factors, while cropping season was considered a random factor due to unpredictable weather conditions. Correlation analysis was performed by Origin 2021 (OriginLab Corp., Northampton, MA, USA) to determine the relationship between GSP composition and soil NO_3_-N content/nitrogen remobilisation/processing qualities. Path analysis was performed by SPSS v.17.0 and the indirect path coefficient was the direct path coefficient multiplied by the correlation coefficient. Graphs were prepared using Origin 2021.

## 3. Results

### 3.1. Effects of Irrigation and Nitrogen Application on GSP and Its Components

Irrigation and nitrogen application had a significant effect on GSP and its components (*p*  < 0.05), and the effects were obviously different in the two soils of differing fertility (Figure 1). Under LF conditions, the impact of irrigation on levels of GSP and its components varied under different nitrogen levels (N0, N180, and N300) for both cultivars. GSP content was significantly higher (*p* < 0.05) in W2 than in W0 under N0 and N180 treatments, and levels were significantly higher (*p* < 0.05) in W0 than in W1 and W2 under N300 treatment. GSP and its components were significantly higher (*p* < 0.05) in W2 than in W0 and W1 under N240 treatment, and glutenin and gliadin levels under W2 treatment were increased by 24% and 13%, and 47% and 16%, for FDC5 and BN207 compared with W0, respectively (Appendix A). Nitrogen application also had a significant influence on GSP levels and components (*p* < 0.01). Under W0 and W1 treatments, GSP levels and components for both cultivars were significantly increased (*p* < 0.05) when nitrogen application was increased from N0 to N300; under W2 treatment, levels increased when nitrogen application increased from N0 to N240, then decreased from N240 to N300, and glutenin and gliadin levels decreased by 14% and 15%, and 22% and 11%, for FDC5 and BN207, respectively (Appendix A).

Importantly, under LF conditions, the highest levels of GSP and its components were obtained for the W2N240 treatment, and glutenin and gliadin levels were increased by 60% and 181%, and 99% and 105%, for FDC5 and BN207, respectively, compared to W0N0 (Appendix A). Under HF conditions, irrigation had different effects on glutenin and gliadin levels in the two cultivars (Figure 1). For FDC5, the highest levels of glutenin and its components were obtained under W1 treatment, and the highest total gliadin and α/β-gliadin levels were obtained under W0 conditions when applying the same nitrogen treatment. For BN207, the glutenin content was significantly increased (*p* < 0.05) when nitrogen application was increased from N0 to N300 under W0 treatment, and the highest gliadin content was achieved under W1 treatment when nitrogen levels were increased from N0 to N240. Under W0 treatment, levels of GSP and its components in both cultivars were increased when nitrogen application was increased from N0 to N300, but glutenin content varied significantly among different nitrogen treatments (*p* < 0.05); the highest glutenin and gliadin contents were obtained under the W1N240, W0N300 treatment and W0N300, W1N240 treatment for FDC5 and BN207, respectively (Figure 1).

GSP content was higher under LF conditions than HF conditions, and irrigation and nitrogen application had different effects on glutenin subunits in the two cultivars. For the strong-gluten cultivar FDC5, Bx7 and Dy10 were increased by 126% and 59%, and 101% and 117%, under LF and HF conditions under optimal treatment compared with W0N0 treatment, respectively. For the medium-gluten cultivar BN207, Ax1 and Dx5 subunits were increased by 154% and 141%, and 183% and 256%, under LF and HF conditions under optimal treatment compared with W0N0 treatment, respectively (Appendix A). Therefore, irrigation and nitrogen application increased glutenins content primarily by increasing Bx7 and Dy10 subunit levels in FDC5, and by enhancing the accumulation of Ax1 and Dx5 subunits in BN207. Under the optimal treatment, on the one hand, LMW-GS and Dx5 levels were 1.3- and 1.2-fold, and 3.1- and 4.1-fold, higher in FDC5 than in BN207 under LF and HF conditions, respectively; and By8/By9 levels were 3.4- and 3.6-fold higher in BN207 than in FDC5 under LF and HF conditions, respectively (Appendix A). This indicates that LMW-GS, Dx5, and By8/By9 levels were the main differences between the two different gluten cultivars, hence they may be the key factors determining variation in glutenin content; on the other hand, α/β-gliadins exhibited the largest responses to irrigation and nitrogen application, increasing by 195% and 104%, and 178% and 221%, for FDC5 and BN207, respectively, under LF and HF conditions under the optimal treatment compared to the W0N0 treatment.

### 3.2. Effects of Irrigation and Nitrogen Application on Processing Qualities

As depicted in Figure 2, irrigation, nitrogen application, and their interaction had significant effects (*p* < 0.01) on processing qualities in both fertility soil types. Under LF conditions, WGC, DST, and SV were increased under W0 and W1 treatments when nitrogen application was increased from N0 to N300, while maximum values were achieved under W2 treatment with nitrogen application at N240, with increases by 98%, 93%, and 89%, and 44%, 27%, and 158%, for FDC5 and BN207 compared with N0 treatment, respectively. Regardless of irrigation, DWR was significantly increased (*p* < 0.05) with increasing nitrogen treatment from N0 to N240. Overall, the highest WGC, DST, DWR, and SV values were obtained under W2N240 treatment, with increases by 114%, 107%, 12%, and 97%, and 65%, 77%, 10%, and 206%, for FDC5 and BN207 compared with W0N0 treatment, respectively (Appendix A). Under HF conditions, under N240 and N300 treatments, W2 treatment significantly reduced (*p* < 0.05) the processing qualities compared with W0 and W1 treatments for both cultivars. Processing qualities were significantly increased (*p* < 0.05) under W0 and W2 treatments when nitrogen application was increased from N0 to N300; under W1 treatment, processing qualities increased when nitrogen application increased from N0 to N240, but there was a significant decline (*p* < 0.05) with a further increase in nitrogen fertiliser from N240 to N300. 

The highest processing qualities were obtained under W1N240 and W0N300 treatments for FDC5 and BN207, respectively, with WGC, DST, DWR, and SV increased by 84%, 79%, 8%, and 84%, and 88%, 124%, 9%, and 93%, for FDC5 and BN207 compared with W0N0 treatment, respectively (Appendix A). It is noteworthy that the enhancing effects of irrigation and nitrogen application on processing qualities were more pronounced for BN207 than FDC5, particularly in terms of SV.

### 3.3. Correlations between GSP and Soil NO_3_-N Content/Nitrogen Remobilisation in Plant Organs

Soil nitrate nitrogen (NO_3_-N) is a crucial nutrient for crop growth, and levels are typically increased by nitrogen application and reduced by irrigation. Correlations between soil NO_3_-N content and GSP levels and components showed disparate results under LF and HF conditions. Under LF conditions, there were significant positive correlations (*p* < 0.05) between soil NO_3_-N content and gliadin composition, with correlation coefficients of 0.685, 0.632, and 0.648, respectively. However, there was no significant correlation between soil NO_3_-N content and glutenin composition (Table 2). Under HF conditions, there was a significant positive correlation (*p* < 0.05) between soil NO_3_-N content and GSP composition, with the highest coefficient of 0.918 between soil NO_3_-N and Dy10. These results indicate that the increased soil NO_3_-N content enhanced GSP levels under HF conditions, with a more pronounced impact on gliadin accumulation under LF conditions.

As shown in Table 2, there was a significant positive correlation (*p* < 0.05) between DNC in stems with sheaths or spikes and Bx7, Dy10 and gliadin components, and between DNC in leaves and GSP components, except for Dx5; coefficients for leaves were higher than those for stems with sheaths and spikes. Dy10 (glutenins) and α/β-gliadin (gliadins) exhibited the strongest correlations with DNC in leaves. This indicates that the increased nitrogen remobilisation in leaves promoted the accumulation of GSP components.

### 3.4. Correlation and Path Analyses of GSPs and Processing Qualities

As shown in Figure 3A, there were significantly positive correlations (*p* < 0.05) between GSP components and processing qualities, except for By8/By9. Specifically, WGC and SV exhibited higher correlations with Bx7, Dy10, α/β-gliadin and γ-gliadin, and DWR showed higher correlations with Ax1, Bx7, α/β-gliadin, and γ-gliadin than for other components. However, there were more pronounced correlations between DST and Dx5, Ax1, and LMW-GS than for other components.

To further explore the influence of individual GSP components on processing qualities, path analysis was conducted, which revealed significant correlations (*p* < 0.05) with processing qualities. Specifically, Dy10 was the factor most strongly influencing WGC among the eight components tested (Figure 3B). Indirect path coefficients showed that LMW-GS, Ax1, Bx7, Dx5, ω-gliadin, α/β-gliadin, and γ-gliadin predominantly influenced WGC through Dy10 (Appendix A). Consequently, elevated Dy10 content emerged as the primary driver of the increase in WGC. For DST, Dx5 was most influential factor. Indirect path coefficients indicated that the components primarily impacted DST through Dx5, except for γ-gliadin. Regarding DWR and SV, γ-gliadin had the strongest direct positive impact, and indirect path coefficients indicated that LMW-GS, Ax1, Bx7, Dy10, ω-gliadin, and α/β-gliadin influenced DWR and SV through γ-gliadin. Overall, increased levels of Dy10, Dx5, and γ-gliadin improved WGC, DST, DWR, and SV, respectively.

## 4. Discussion

### 4.1. Irrigation and Nitrogen Significantly Affect GSP Content and Processing Qualities

Glutenin and gliadin components are associated with the processing and nutritional quality of wheat, and they are affected by environmental factors such as irrigation and fertilisation [19]. Most previous studies have focused on the individual effects of irrigation or nitrogen application treatment on the GSP content of wheat, while the interaction effects on GSP composition have been neglected. Previous studies indicate that irrigation during the late growing period significantly decreases the protein content and shortens the dough stability time, ultimately resulting in lower wheat quality [42]. With increasing irrigation, dough stability time and dough water absorption rate exhibited parabolic trends by first increasing, reaching peak values under conditions with two irrigation steps (applied at wintering and jointing stages), then decreasing with increasing irrigation under conditions with three irrigation steps (applied at wintering, jointing, and flowering stages) [20]. In the present study, GSP content, wet gluten content, and dough stability time were significantly decreased at W2 compared with W1 treatment under HF conditions, and higher GSP content and processing qualities were achieved under W0 or W1 conditions. It indicated that less irrigation in wheat late growth stage might achieve better processing quality. The result aligned with previous findings where no irrigation during the flowering stage improved dough stability time and sedimentation value [43]. 

Nitrogen application affects protein content and composition, in turn influencing the mixing properties of dough and wheat quality [15]. Protein content, wet gluten content, and dough stability time were increased with an increasing nitrogen application rate within the range of 0–240 kg/ha, but protein content and processing qualities tended to decrease when the nitrogen application rate exceeded 240 kg/ha [44], consistent with our results under irrigation at two locations. It was contributed by dilution effects of the increased grain yield on grain protein content [45]. However, under water-limited conditions, the GSP content and processing qualities were increased with the increasing of nitrogen application within the range of 0–300 kg/ha, because the limited irrigation affected the effectiveness of nitrogen fertiliser [46]. Also, some studies have indicated that nitrogen rates exceeding 168 kg/ha yielded minimal further improvement in flour quality [47]. These apparent discrepancies may reflect variations in soil fertility and differences in characteristics between plant cultivars.

Previous reports also found that nitrogen application could influence the effect of irrigation on protein content; for example, irrigation in the range of 0–225 mm had a diluting effect on grain quality, while protein content and wet gluten content were decreased with increasing irrigation, and this effect could be diminished by nitrogen application [31]. Moderately high nitrogen fertilisation under water deficit conditions not only alleviated losses of grain yield caused by water deficit, but also increased protein accumulation [15,48]. Our current work showed that high nitrogen fertilisation significantly increased GSP content under no irrigation conditions in HF conditions, but it decreased significantly under two irrigation steps. This is mainly due to the dilution effects of irrigation on grain protein content [49,50]. We also found that an application rate of 240 kg/ha effectively mitigated the reduction in wet gluten content, dough stability time, and sedimentation value under irrigation treatments. Liu et al. also found that nitrogen application of 240 kg/ha under irrigation increased the content of β-sheets and random coils of gluten protein and improved the gluten secondary structure, resulting in improvement of dough rheological properties [15]. These results are consistent with those of previous studies. N0 fertiliser treatments produced the lowest GSP content and the poorest processing qualities at all irrigation levels. Irrigation was able to increase both GSP content and processing qualities under nitrogen application conditions [51]. However, irrigation could not balance the adverse effect of nitrogen stress on GSP content and processing qualities, although it did indeed increase these parameters.

### 4.2. Dx5, Dy10 and γ-Gliadin Are Direct Contributors to Processing Qualities

WGC, DST, DWR, and SV are intimately related to the quality of bread and steamed bread [52,53]. In the case of strong-gluten wheat, which is suitable for making bread, higher WGC, SV, and longer DST facilitate the formation of the internal structure of the dough and expansion of the bread bulk, as well as better formation of the honeycomb texture [54]. For medium-gluten wheat, which is more applicable for making steamed breads, DST, DWR, and SV are significantly positively correlated with steamed bread size, height, specific volume, and rating score [55]. Therefore, it is worthwhile to investigate the effect of differences in storage protein composition on wheat processing qualities. Previous studies have demonstrated that utilising the HMW-GS content of wild relatives could improve the quality of wheat for breadmaking. The Glu-D locus was found to be an important determinant of breadmaking quality [56,57]. An increase in HMW-GS content can enhance dough quality [58]. The Ax1, Dx5, and Dy10 subunits have a positive effect on steamed bread size [59,60]. Our study showed that DST and WGC were improved by Dx5 and Dy10, respectively, and probably further influenced the quality of bread and steamed bread, indicating that Dx5 and Dy10 subunits played a crucial role in improving wheat qualities. There was also a considerable indirect influence between glutenin composition and processing qualities; for example, LMW-GS, Ax1, Bx7, and Dx5 affected WGC predominantly through Dy10, whereas LMW-GS, Ax1, Bx7, and Dy10 affected DST through Dx5. Correlation analysis revealed a negative correlation between By8/By9 and processing qualities. However, previous studies showed that Bx7 + By8 or Bx7 + By9 were associated with higher dough strength and better breadmaking performance [61,62]. The possible reason for this discrepancy might be the greater abundance of cysteines in Bx7, since they play a crucial role in the formation of HMW-GS polymers, and thereby influence dough qualities, but By8 or By9 alone may not be conducive to improving dough qualities.

The roles of gliadin in breadmaking quality remain contentious. Some previous studies reported that the gliadin content was not correlated with any of the rheological properties of dough and gluten [63]. By contrast, other researchers found that gliadins were directly related to dough rheological properties and bread-loaf volume [64,65]. γ-gliadin was an important component in GSP to improve the wheat qualities. The current work showed that γ-gliadin affected SV and may further influence the processing qualities of bread and steamed bread, consistent with previous studies [66]. Additionally, γ-gliadin was also a direct factor affecting DWR. However, further studies are needed to unveil the molecular mechanisms of individual storage protein components in determining the processing qualities of bread and steamed bread.

### 4.3. Irrigation and Nitrogen Affect Yield and Grain Storage Proteins Differently in Different Soil Fertility Types

Grain yield and grain protein content are important traits for wheat production, but they are usually negatively correlated [67,68,69]. This is related to the fact that protein synthesis requires more energy and photosynthetic products than carbohydrate synthesis, or it may be attributed to the dilution effect of carbohydrates on grain protein [70]. However, some researchers found that protein content can display positive and negative correlations with yield under drought stress and irrigated conditions, respectively [71]. In the present study, grain yield, GSP yield, and GSP content were maximised under W2N240 treatment and LF conditions for both cultivars (Appendix A), attributed to the transfer of nitrogen from vegetative organs to grains [72]. Two irrigation steps increased the effectiveness of nitrogen fertiliser [73], resulting in a combined growth in yield and storage protein content, but grain yield and grain protein content were not negatively correlated, in contrast to the results of previous studies [67,68,69]. This may be due to the lower soil nitrogen content, the lower moisture content, or the poorer fertiliser conservation capacity in low fertility soils. Under HF conditions, maximum yields were achieved under W2N180 treatment for both cultivars; the 2-year average yields were 2.4- and 2.3-fold higher than W0N0 treatments for FDC5 and BN207, respectively (Appendix A). This indicated that fertile soil had relative higher capacity to supply nitrogen, and less nitrogen application was needed for high yield under irrigation conditions. However, the highest GSP content was obtained under W1N240 and W0N300 treatments for FDC5 and BN207, respectively, and the highest GSP yields were obtained under W1N240 treatment (Appendix A). This suggested that the limited irrigation affected the effectiveness of nitrogen fertiliser, and more nitrogen application was needed for high GSP content. These results are consistent with the negative correlation between protein content and grain yield reported in previous studies [67,68,69]. Moreover, our study found that GSP content was higher under LF conditions than HF conditions. This could be due to the dilution effect by the higher yield under HF conditions, resulting in lower GSP content under HF conditions than LF conditions. Therefore, strategies involving irrigation and nitrogen for high yield and quality of wheat should consider both soil types and fertility levels.

## 5. Conclusions

Our research revealed that appropriate irrigation and nitrogen application can increase soil NO_3_-N content, and the higher soil NO_3_-N content increased nitrogen remobilisation in leaves, which in turn increased the GSP content. Irrigation and nitrogen application increased GSP levels mainly by enhancing the accumulation of Ax1, Bx7, Dx5, Dy10, and α/β-gliadin. Dx5, Dy10, and γ-gliadin were the key factors influencing dough stability time, wet glutenin content, dough water absorption rate, and sedimentation value, respectively. High GSP yields and processing qualities were obtained under a nitrogen application rate of 240 kg/ha with a single irrigation under HF conditions, and the same nitrogen application rate with two irrigation steps under LF conditions (Figure 4).

## Figures and Tables

**Figure 1 foods-12-04407-f001:**
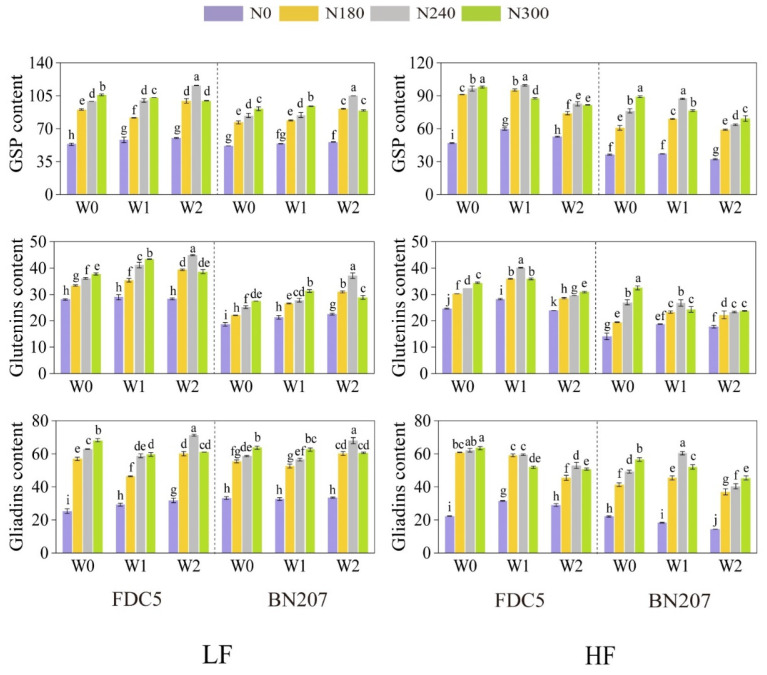
Levels of GSP and its components under different irrigation and nitrogen application treatments in two different soil fertility types; 10^6^ AU/mg, AU represents the peak area of each fraction. LF, low soil fertility; HF, high soil fertility; FDC5, Fengdecun5; BN207, Bainong207; LMW-GS, low-molecular-weight glutenin subunits. Different lowercase letters indicate significant differences among different treatments within each cultivar (*p* < 0.05).

**Figure 2 foods-12-04407-f002:**
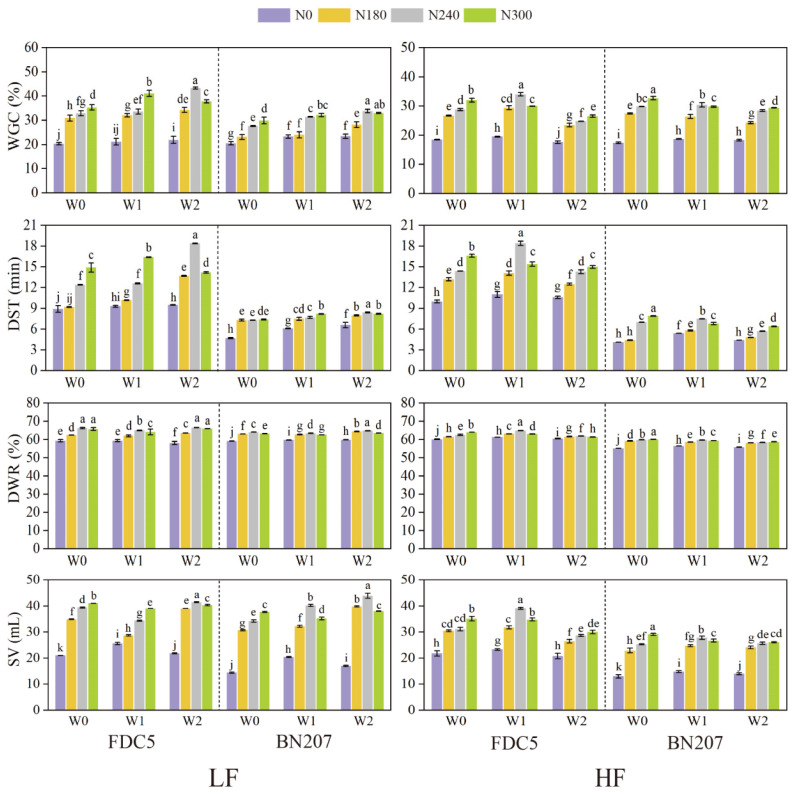
Processing quality parameters under different irrigation and nitrogen application treatments. Values are annual averages for 2018–2019 and 2019–2020. LF, low soil fertility; HF, high soil fertility; FDC5, Fengdecun5; BN207, Bainong207; WGC, wet gluten content; DST, dough stability time; DWR, dough water absorption rate; SV, sedimentation value. Different lowercase letters indicate significant differences among different treatments within each cultivar (*p* < 0.05).

**Figure 3 foods-12-04407-f003:**
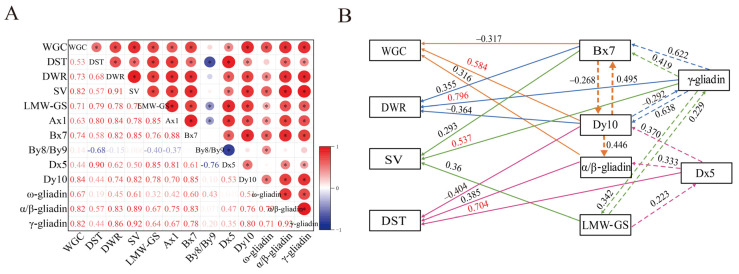
Correlation and path analyses of GSP and processing qualities. (**A**) Correlation coefficients between GSP and processing qualities. (**B**) Path analysis of GSP and processing qualities. Analyses were conducted based on the two fertilities soil types as replicates. * Represents significance at *p* < 0.05. Orange lines represent relationships with WGC. Blue lines represent relationships with DWR. Green lines represent relationships with SV. Red lines represent relationships with DST. Straight lines represent direct relationships with processing qualities. Dashed lines represent indirect relationships with processing qualities. Part (**B**) only shows the three greatest direct and indirect impact factors. The complete results from path analysis are included in Appendix A. LMW-GS, low-molecular-weight glutenin subunits; WGC, wet gluten content; DST, dough stability time; DWR, dough water absorption rate; SV, sedimentation value.

**Figure 4 foods-12-04407-f004:**
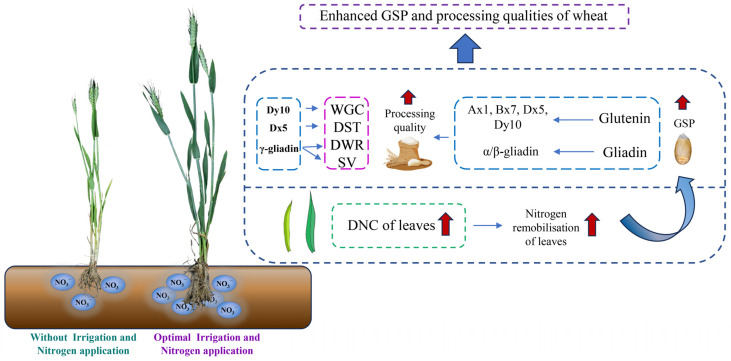
Appropriate irrigation and nitrogen application improves the GSP content and processing qualities of wheat. GSP, grain storage proteins; LMW-GS, low-molecular-weight glutenin subunits; WGC, wet gluten content; DST, dough stability time; DWR, dough water absorption rate; SV, sedimentation value; DNC, differences in nitrogen concentration in vegetative organs between flowering and maturity stages.

**Table 1 foods-12-04407-t001:** Physicochemical characteristics of the 0–20 cm soil layer.

Soil Fertility	Soil Texture	Organic Matter	Available Phosphorus	Available Potassium	NO_3_-N	Total N
		(g·kg^−1^)	(mg·kg^−1^)	(mg·kg^−1^)	(mg·kg^−1^)	(g·kg^−1^)
LF	Sandy	8.33	7.43	94.32	8.1	0.53
HF	Clay	18.51	15.51	170.64	25.9	0.98

**Table 2 foods-12-04407-t002:** Correlations between GSP composition and soil NO_3_-N content/nitrogen remobilisation in plant organs.

Traits	NO_3_-N		DNC		
	LF	HF	Leaves	Stems with Sheaths	Spikes
Glutenins	0.519	0.762 **	0.418 **	0.256	0.271
LMW-GS	0.492	0.656 *	0.364 **	0.215	0.217
Ax1	0.464	0.743 **	0.350 *	0.212	0.27
Bx7	0.413	0.903 **	0.531 **	0.320*	0.359 *
By8/By9	0.535	0.787 **	0.428 **	0.215	0.265
Dx5	0.436	0.812 **	0.064	0.078	0.075
Dy10	0.455	0.918 **	0.623 **	0.409 **	0.438 **
Gliadins	0.660 *	0.873 **	0.650 **	0.578 **	0.531 **
ω-gliadin	0.685 *	0.895 **	0.432 **	0.546 **	0.306 *
α/β-gliadin	0.632 *	0.884 **	0.631 **	0.572 **	0.527 **
γ-gliadin	0.648 *	0.863 **	0.585 **	0.561 **	0.454 **

* and ** represent significance at *p* < 0.05 and *p* < 0.01, respectively. NO_3_-N content in the 0–20 cm soil layers is shown in Appendix A. DNC in plant organs is shown in Appendix A with annual average values for 2018–2019 and 2019–2020. LF, low soil fertility; HF, high soil fertility; LMW-GS, low-molecular-weight glutenin subunits.

## Data Availability

The authors confirm that the data supporting the findings of this study are available within the article.

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
