# Peer review of "Wheat GSPs and Processing Quality Are Affected by Irrigation and Nitrogen through Nitrogen Remobilisation"

_foods, 2023, doi:10.3390/foods12244407_

Round 1
Reviewer 1 Report
Comments and Suggestions for Authors
Dear Authors,
The manuscript presents an interesting topic but the presentation of results especially the figures should be improved.
Below are specific comments
Review comments-Foods
The manuscript presents interesting topic but the results need to be rewritten for simplicity. Here are some comments below to be addressed.
Tittle;
· “Wheat Grain Storage Proteins and Processing Quality are affected by Irrigation and Nitrogen Application through Nitrogen Remobilisation”
· The tittle is too long, should be shortened to capture key variables and be concise/informative.
Abstract;
· Authors should rewrite N180, 180 kg·hm-2; N240, 240 kg·hm-2 ; N300, 300 kg·hm-2 ), in conventional, widely used units, kg/ha.
· Include percentage changes compared to control while presenting key results.
· The word significant should be accompanied by P ≤ 0.05
Introduction;
· Authors should add literature highlighting different levels of N and irrigation on GSP in other crops or in Wheat. The introduction does not provide detailed research gap in its current state.
Methodology
· The units kg/ha should be applied to the rates of nitrogen
· The statistical analysis performed is not clear;
· Please provide how interaction s between levels of N and irrigation were determined. Did you do a two-away ANOVA, or three way (Nitrogen x Irrigation x Cultivar). It is a must that a split plot design be accompanied by interactions found by either 2- or 3-way ANOVA.
Results;
· Figure 1 is too congested to be followed by readers. It does not perform its work of presentation of results. We use Figures for showing trends and Tables for comparisons. Authors should carefully think of which data should be presented in tables and which on figures (Not more than 6 graphs). The tables should clearly indicate the independent variables and dependent variables of interest for clear understanding of this work. Otherwise figures 1 and 2 are too congested, only understood by the authors and not readers.
· Description of results should clearly indicate the levels of statistical significance i.e. p ≤ 0.05, or p ≤ 0.01
· There should be a table of ANOVA, where 2-way ANOVA 9sources of variation, treatment and treatment interactions are clearly shown)
Discussion;
· The discussion section should be improved
· Only mentioning that our results are consistent with….x et al. is not sufficient in the discussion section. Results should be interpreted, i.e. derive meaning or implications and highlight the novelty of the key findings. Also, provide detailed explanation of the mechanisms underlying the observations. Moreover, the interactions between fertility and irrigation has not been well discussed in section 4.1.
Comments on the Quality of English Language
The English is moderate
Author Response
Responses to the comments
Dear Editor,
Many thanks for you and the reviewers’ comments for revision of the manuscript - Foods-2724388.
All the changes in the revised version were marked as red for easy recognizing.
Detailed responses are listed as below.
Best Wishes
Sincerely Yours
Yuanxin Shen
- Tittle
“Wheat Grain Storage Proteins and Processing Quality are affected by Irrigation and Nitrogen Application through Nitrogen Remobilisation” The tittle is too long, should be shortened to capture key variables and be concise/informative.
Response: Accepted the suggestions. In the revised manuscript, the tittle was rewriting as “Wheat GSPs and Processing Quality are Affected by Irrigation and Nitrogen through Nitrogen Remobilisation”.
- Abstract
Authors should rewrite N180, 180 kg·hm-2; N240, 240 kg·hm-2; N300, 300 kg·hm-2), in conventional, widely used units, kg/ha.
Response: We corrected the units throughout the manuscript.
Include percentage changes compared to control while presenting key results.
Response: We have added the informations about the percentage changes compared to control in the abstract
The word significant should be accompanied by P ≤ 0.05
Response: We agreed with the comment, and checked it throughout the manuscript.
- Introduction
Authors should add literature highlighting different levels of N and irrigation on GSP in other crops or in Wheat. The introduction does not provide detailed research gap in its current state.
Response: We accepted the suggestion. In the introduction section in the revised manuscript, we added more information on different levels of nitrogen and irrigation on GSP in Wheat. Please see References [15], [16], [17,18], [19], [20,21], [25], [26], [27], [28], [29,30], [29], and [31] in the introduction section.
- Methodology
The unit’s kg/ha should be applied to the rates of nitrogen
Response: We have corrected the units throughout the manuscript according to the suggestion.
The statistical analysis performed is not clear; Please provide how interaction s between levels of N and irrigation were determined. Did you do a two-away ANOVA, or three ways (Nitrogen x Irrigation x Cultivar). It is a must that a split plot design be accompanied by interactions found by either 2- or 3-way ANOVA.
Response: We have added a two-way MIXED analysis of variance (ANOVA) to examine the interactions between irrigation and nitrogen application on the GSP and its components and the processing qualities in materials and methods.
- Results
Figure 1 is too congested to be followed by readers. It does not perform its work of presentation of results. We use Figures for showing trends and Tables for comparisons. Authors should carefully think of which data should be presented in tables and which on figures (Not more than 6 graphs). The tables should clearly indicate the independent variables and dependent variables of interest for clear understanding of this work. Otherwise figures 1 and 2 are too congested, only understood by the authors and not readers.
Response: We have modified the Figure 1 and made it simply. And it only showed the effects of different irrigation times and nitrogen application on GSP, glutenins, and gliadins contents in the two cultivars under low and high soil fertility conditions. The remaining data of the GSP components have been placed in Table S1 and Table S2 in supplementary information, which correspond to the percentage increase or decrease dates mentioned in the results.
Description of results should clearly indicate the levels of statistical significance i.e. p ≤ 0.05, or p ≤ 0.01
Response: We corrected it and checked throughout the manuscript.
There should be a table of ANOVA, where 2-way ANOVA 9sources of variation, treatment and treatment interactions are clearly shown)
Response: We have revised this section in the supplementary information. The additional informations in Tables S1, S2, and S3 include the effects of individual irrigation or nitrogen application, as well as the analysis of variance of their interaction on GSP compositions and processing qualities.
- Discussion
The discussion section should be improved. Only mentioning that our results are consistent with….x et al. is not sufficient in the discussion section. Results should be interpreted, i.e. derive meaning or implications and highlight the novelty of the key findings. Also, provide detailed explanation of the mechanisms underlying the observations. Moreover, the interactions between fertility and irrigation has not been well discussed in section 4.1.
Response: We accepted the suggestion and revised this part. We have improved the interpretation of the results as much as possible through literature support. Please see References [43], [45], [46], [49,50], and [15] in the discussion section.
The interactions between fertility and irrigation have been added in discussion 4.1.

Reviewer 2 Report
Comments and Suggestions for Authors
Bread wheat (Triticum aestivum) is characterized by high content a storage proteins in the grain, and its value is determined by the content and composition of the storage proteins. Among the technological methods for increasing the productivity of wheat and other cereals, the use of nitrogen fertilizers should be noted, since almost all available nitrogen, which is the basis for plant growth and development, is annually removed from the soil with the harvest. In this regard, the article by respected authors, devoted to the study of the combined influence of irrigation and nitrogen on the composition of storage proteins in wheat, is relevant and is of undoubted scientific and practical interest.
The article is interesting, original, supplemented with tables and very good illustrations, which is undoubtedly a big plus. I would also like to note the undoubted advantage of the article - a detailed description of “materials and methods”, as well as a succinct “discussion” and “conclusion”. I believe that the article by respected authors will be of interest to a wide range of scientists related to crop production.
Author Response
We greatly appreciate the time and effort you dedicated to reviewing our manuscript. and we are grateful for your positive recommendation.
Reviewer 3 Report
Comments and Suggestions for Authors
The manuscript entitled "Wheat Grain Storage Proteins and Processing Quality are Affected by Irrigation and Nitrogen Application through Nitrogen Remobilisation" addresses the topic of irrigation and nitrogen fertilization role on the qualitative aspects of the most important crop for food in the world.
I found the study needs minor revisions.
Having carefully reviewed the manuscript, I would like to offer some constructive feedback and suggestion for improvement that I believe would enhance the impact and the international appeal of the work.
I appreciated the study, which is focused on the specific and technological aspects of wheat. However, I believe it should be highlighted that the effect of climate change at a global level is increasing uncertainty in the field of crop management from an agronomic point of view. This involves a progressive increase in the water needs of crops or the need to stabilize quantitative-qualitative yields through the adoption of supplemental irrigation. This scenario concerns many agricultural crops that are worldwide crucial for human nutrition and concerns many geographical areas of the planet.
Additional information can be found in Campesi et al., 2023 (DOI: 10.31545/intagr/162340)
I will appreciate if some sentences will be added in the introduction section capable of extend the potentiality of your research to other environments and crop systems, as well as allowing to consider the results of the study as an evaluation paper of a crop management adaptation strategy to climate change.
Author Response
We would like to thank the reviewer for the thorough review of our manuscript and the positive comments. In addition, I referred to the literature according to the reviewer’s suggestion (Campesi et al., 2023) and quoted some sentences about the global climate changes and the impacts of supplementary irrigation on crop growth in the introduction section. Please see References [11] and [11,14] in the introduction section.